# Workflow Graphs: A Computational Model of Collective Task Strategies for 3D Design Software

Minsuk Chang*
School of Computing
KAIST

Ben Lafreniere†
Autodesk Research

Juho Kim‡
School of Computing
KAIST

George Fitzmaurice§
Autodesk Research

Tovi Grossman¶
University of Toronto

## ABSTRACT

This paper introduces *Workflow graphs*, or *W-graphs*, which encode how the approaches taken by multiple users performing a fixed 3D design task converge and diverge from one another. The graph's nodes represent equivalent intermediate task states across users, and directed edges represent how a user moved between these states, inferred from screen recording videos, command log data, and task content history. The result is a data structure that captures alternative methods for performing sub-tasks (e.g., modeling the legs of a chair) and alternative strategies of the overall task. As a case study, we describe and exemplify a computational pipeline for building *W-graphs* using screen recordings, command logs, and 3D model snapshots from an instrumented version of the Tinkercad 3D modeling application, and present graphs built for two sample tasks. We also illustrate how *W-graphs* can facilitate novel user interfaces with scenarios in workflow feedback, on-demand task guidance, and instructor dashboards.

**Index Terms:** Human-centered computing—Interactive systems and tools—;—

## 1 INTRODUCTION

There are common situations in which many users of complex software perform the same task, such as designing a chair or table, bringing their unique set of skills and knowledge to bear on a set goal. For example, this occurs when multiple people perform the same tutorial, complete an assignment for a course, or work on sub-tasks that frequently occur in the context of a larger task, such as 3D modeling joints when designing furniture. It is also common for users to discuss and compare different methods of completing a single task in online communities for 3D modeling software (for an example of such discussion, see Figure 2). This raises an interesting possibility—what if the range of different methods for performing a task could be captured and represented as rich workflow recordings, as a way to help experienced users discover alternative methods and expand their workflow knowledge, or to assist novice users in learning advanced practices?

In this research, we investigate how multiple demonstrations of a fixed task can be captured and represented in a *workflow graph (W-graph)* (Figure 1). The idea is to automatically discover the different means of accomplishing a goal from the interaction traces of multiple users, and to encode these in a graph representation. The graph thus represents diverse understanding of the task, opening up a range of possible applications. For example, the graph could be used to provide targeted suggestions of segments of the task for which alternative methods exist, or to synthesize the most efficient means

*e-mail: minsuk@kaist.ac.kr
†e-mail: ben.lafreniere@gmail.com
‡e-mail: juhokim@kaist.ac.kr
§e-mail: george.fitzmaurice@autodesk.com
¶e-mail: tovi@dgp.toronto.edu

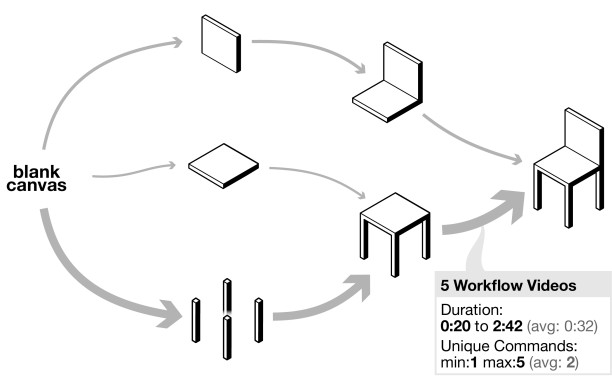

Figure 1: W-graphs encode multiple demonstrations of a fixed task, based on commonalities in the workflows employed by users. Nodes represent semantically similar states across demonstrations. Edges represent alternative workflows for sub-tasks. The width of edges represents the number of distinct workflows between two states.

of completing the task from the many demonstrations encoded in the graph. It could also be used to synthesize and populate tutorials tailored to particular users, for example by only showing methods that use tools known to that user.

To investigate this approach, we instrumented *Tinkercad*[1], a 3D solid modeling application popular in the maker community, to gather screen recordings, command sequences, and changes to the CSG (constructive solid geometry) tree of the specific 3D model being built. The interaction traces for multiple users performing the same task are processed by an algorithm we developed, which combines them into a W-graph representing the collective actions of all users. Unlike past approaches to workflow modeling in this domain, which have focused on command sequence data (e.g., [38]), our approach additionally leverages the 3D model content being created by the user. This allows us to track the progress of the task in direct relation to changes in the content (i.e., the 3D model) to detect common stages of the task progression across multiple demonstrations. We use an autoencoder [34] to represent the 3D geometry information of each 3D model snapshot, which we found to be a robust and scalable method for detecting workflow-relevant changes in the geometry, as compared to metrics such as comparing CSG trees, 2D renders, and 3D meshes.

The result is a graph in which each directed edge from the starting node to a terminal node represents a potential workflow for completing the task, and multiple edges between any two states represent alternative approaches for performing that segment of the task. The collected command log data and screen recordings associated with the edges of the graph can be processed to define metrics on paths (such as average workflow duration or number of unique commands used), and displayed as demonstration content in interfaces.

The main contributions of this paper are:

---

[1]https://tinkercad.com

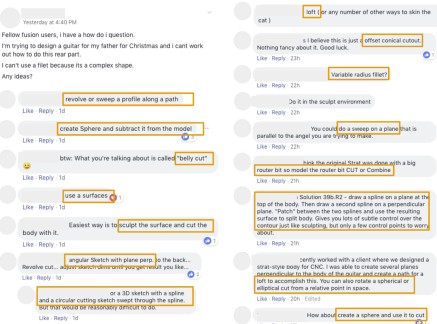

Figure 2: Fifteen distinct suggestions on how to perform a 3D modeling task – from the largest Fusion 360 user community on Facebook

- The concept of W-graphs, which represent the semantic structure of a task, based on demonstrations from multiple users
- A computational pipeline for constructing W-graphs and a demonstration of the approach for sample tasks in Tinkercad
- The description of possible applications enabled by W-graphs

We begin with a review of prior work, then describe the W-graph construction approach at a conceptual level. Next, we present workflow graphs constructed for two sample tasks performed by Tinkercad users, and discuss three applications enabled by W-graph—workflow feedback, on-demand task guidance, and instructor support. Finally, we present preliminary user feedback on a prototype of one of these applications, W-suggest, and conclude with a discussion of directions for future work.

## 2 RELATED WORK

This work expands prior research on software learning and workflow capture, mining organically-created instructional content, and supporting learning at scale.

### 2.1 Software Learning and Workflow Capture

Early HCI research recognized the challenges of learning software applications [7], and identified the benefits of minimalist and task-centric help-resources [6]. More recently, Grossman et al. [21] identified five common classes of challenges that users face when learning feature-rich software applications: understanding how to perform a task, awareness of tools and features, locating tools and features, understanding how to use specific tools, and transitioning to efficient behaviors.

Of the challenges listed above, the majority of existing work on assisting users to acquire alternative workflows has looked at how to promote the use of keyboard shortcuts and other expert interaction techniques [20,30,31,36], with less attention on the adoption of more efficient workflows. Closer to the current work is CADament [29], a real-time multi-player game in which users compete to try and perform a 2D CAD task faster than one another. In the time between rounds of the game, the user is shown video of peers who are at a higher level of performance than they are, a feature which was found to prompt users to adopt more efficient methods. While CADament shares some similarity with the current work, the improvements were at the level of refining use of individual commands, rather than understanding alternative multi-command workflows.

Beyond systems explicitly designed to promote use of more efficient behaviors, a number of systems have been designed to capture workflows from users, which could then be made available to others. Photo Manipulation Tutorials by Demonstration [19] and MixT [10] enable users to perform a workflow, and automatically convert that demonstration into a tutorial that can be shared with other users. Meshflow [12] and Chronicle [22] continuously record the user as they work, capturing rich metadata and screen recordings, and then

provide visualizations and interaction techniques for exploring that editing history. In contrast to these works, which capture individual demonstrations of a task, W-graphs captures demonstrations from multiple users, and then uses these to recommend alternate workflows. In this respect, the current work is somewhat similar to Community Enhanced Tutorials [28], which records video demonstrations of the actions performed on each step of an image-editing tutorial and provides these examples to subsequent users of the tutorial. However, W-graphs looks at a more general problem, where the task is not sub-divided into pre-defined steps, and users thus have much more freedom in how they complete the task.

Summarizing the above, there has been relatively little work on software learning systems that capture alternative workflows, and we are unaware of any work that has tried to do so by building a representation that encompasses many different means of performing a fixed 3D modeling task.

### 2.2 Mining and Summarizing Procedural Content

A number of research projects have investigated how user-created procedural content can be analyzed or mined for useful information. RecipeScape [9] enables users to browse and analyze hundreds of cooking instructions for an individual dish by visually summarizing their structural patterns. Closer to our domain of interest, Delta [27] produces visual summaries of image editing workflows for Photoshop, and enables users to visually compare pairs of workflows. We take inspiration from the Delta system and this work's findings on how users compare workflows. That being said, our focus is on automatically building a data structure representing the many different ways that a task can be performed, rather than on how to best visualize or compare workflows.

Query-Feature Graphs [16] provide a mapping between high-level descriptions of user goals and the specific features of an interactive system relevant to achieving those goals, and are produced by combining a range of data sources, including search query logs, search engine results, and web page content. While this approach could be valuable for understanding the tasks performed in an application, and the commands related to those commands, query-feature graphs do not in themselves provide a means of discovering alternative or improved workflows.

Several research projects have investigated how to model a user's context as they work in a software application with the goal of aiding the retrieval and use of procedural learning content, for example using command log data [32], interactions gathered through accessibility APIs across multiple applications [17], or coordinated web browser and application activities [15]. Along similar lines, Wang et al. [38] developed a set of recommender algorithms for software workflows, and demonstrated how they could be used to recommend community-generated videos for a 3D modeling tool. While the above works share our goal of providing users with relevant workflow information, their algorithms have focused on using the stream of actions being performed by the user, not the content that is being edited. Moreover, these techniques are not designed to capture the many different ways a fixed task can be performed, which limits their ability to recommend ways that a user can improve on the workflows they already use.

### 2.3 Learning at Scale

A final area of related work concerns how technology can enable learning at scale, for example by helping a scarce pool of experts to efficiently teach many learners, or by enabling learners to help one another. As a recent example, CodeOpticon [23] enables a single tutor to monitor and chat with many remote students working on programming exercises through a dashboard that shows each learner's code editor, and provides real-time text differences in visualizations and highlighting of compilation errors.

Most related to the current work are *learnersourcing* techniques, which harness the activities of learners to contribute to human computation workflows. This approach has been used to provide labeling of how-to videos [25], and to generate hints to learners by asking other learners to reflect on obstacles they have overcome [18]. The AXIS system [40] asks learners to provide explanations as they solve math problems, and uses machine learning to dynamically determine which explanations to present to future learners.

Along similar lines, Whitehill and Seltzer investigated the viability of crowdsourcing as a means of collecting video demonstrations of mathematical problem solving [39]. To analyze the diversity of problem-solving methods, the authors manually extracted the problem solving steps from 17 videos to created a graph of different solution paths. W-graphs produce a similar artifact for the domain of software workflows, and with an automated approach for constructing the graphs.

In summary, by capturing and representing the workflows employed by users with varying backgrounds and skill levels, we see W-graphs as a potentially valuable approach for scaling the learning and improvement of software workflows.

## 3 WORKFLOW GRAPHS

The key problem that we address is that designers and researchers currently lack scalable approaches for analyzing and supporting user workflows. To develop such an approach, we need techniques that can map higher level user intents (e.g., 3D modeling a mug), to strategy level workflows (e.g., modeling the handle before the body), and user actions (the specific sequence of actions involved).

We can broadly classify approaches for modeling user workflows derived from action sequences into bottom-up approaches and top-down approaches.

Bottom-up approaches record users' action sequences, and then attempt to infer the user's intent at a post-processing stage using unsupervised modeling techniques such as semantic segmentation, clustering, or topic modeling [2, 9]. A disadvantage of this approach is that the results can be difficult to present to users, because the results of unsupervised modeling techniques are not human-readable labels. Meaningful labels could conceivably be added to the resulting clusters (e.g., using crowdsourcing techniques [11, 26, 37]), but this is a non-trivial problem under active research.

An alternative is a top-down approach, in which a small number of domain experts break down a task into meaningful units (e.g., subgoals [8]), and then users or crowdworkers use these pre-created units as labels for their own command log data, or that of other users. This approach also comes with disadvantages—users must perform the labeling, their interpretation of pre-defined labels can differ, and the overall breakdown of the task depends on the judgement of a few domain experts, limiting the scalability of the approach.

Then, how can we develop an approach for organizing users' collective interaction data into a meaningful structure while maintaining the scalability of naively recording user action sequences without interrupting them to acquire any labels?

To investigate this possibility, we developed Workflow graphs (W-graphs), which synthesize many demonstrations of a fixed task (i.e., re-creating the same 3D model) such that the commonalities and differences between the approaches taken by users are encoded in the graph. To ensure the technique can scale, the goal is to automate the construction process, using recordings of demonstrations of the task as input (which may include screen recordings, command log data, content snapshots, etc.).

Formally, a W-graph is a directed graph $G = (V, A)$ which consists of the following components:

### 3.1 Graph Vertices

$$V = \{v_i; 1 \leq i \leq N\}$$

The vertices of the graph represent semantically-meaningful states in the demonstrations, such as a sub-goal of the task. These states can be thought of as sub-goals in the workflow—ideally, we want them to capture the points where a user has completed a given sub-task, and has yet to start the next sub-task. Detecting these states automatically from unlabeled demonstrations is a challenge, but the idea is to leverage the demonstrations of multiple users to discover common states that occur across their respective methods for completing the task. If a new demonstration is completely different from those already represented in the graph, it might not share any nodes with those already in the graph, apart from the start and final nodes, which are shared by all demonstrations.

Note that the appropriate criteria for judging which states from multiple demonstrations are semantically-similar is ill-defined, and dependent on the intended application of the W-graph. For example, one criterion could be used to construct a W-graph that indicates coarse differences between approaches for completing the task, while a more strict criterion for similarity could create a more complex graph, which reveals finer differences between similar approaches. As we discuss in the next section, our algorithm allows the threshold for the similarity to be tuned based on the intended application.

### 3.2 Graph Edges

$$A = \{(v_i, v_j, d_k, E_{i,j}); v_i, v_j \in V\}$$

$$E_{i,j,k} = \{event_1, event_2, event_3, \ldots\}$$

The directed edges of the graph represent workflows used by a user to move between semantically-similar states. There may be multiple directed edges between a given pair of states, if multiple demonstrations $d_k$ include a segment from state $v_i$ to $v_j$.

Each directed edge is associated with a set of events $E_{i,j,k}$ which include the timestamped interaction trace of events in demonstration $d_k$ performed in the segment between state $v_i$ and $v_j$. This trace of events could includes timestamped command invocations, 3D model snapshots, or any other timestamped data that was gathered from the recorded demonstrations.

### 3.3 Interaction Data

The interaction trace data associated with edges enables a great deal of flexibility in how the W-graph is used. For example, this data could be used to retrieve snippets of screen recordings of the demonstrations associated with the segment of the task between two states, or it could be used to define metrics on the different workflows used for that segment of the task (e.g., the number of unique commands used, or the average time it takes to perform the workflow). As another example, analyzing the interaction traces along many different paths between states can reveal the average time for sub-tasks or the variance across users. Later in the paper, we present some example applications of W-graphs to illustrate the full flexibility of this data representation.

## 4 PIPELINE FOR CONSTRUCTING W-GRAPHS

In this section we describe the computational pipeline we have developed for constructing W-graphs. We start by discussing our instrumentation of Tinkercad and the data set we collected, then present the multi-step pipeline for processing the data and the similarity metric for identifying equivalent-intermediate states. The choice of a method for identifying equivalent-intermediate states is a key aspect of the pipeline, and we experimented with several alternative methods.

### 4.1 Tinkercad Data Collection

We instrumented a customized version of Tinkercad to record time-stamped command invocations and snapshots of the 3D model the user is working on after each command is executed (represented as a constructive solid geometry (CSG) tree with unique IDs for each object, to enable the association of model parts across multiple snapshots). To capture the instrumentation data, participants were asked to install Autodesk Screencast[2], a screen recording application that can associate command metadata with the timeline of recorded video data. Collectively, this allowed us to gather timestamp-aligned command invocation data, 3D model snapshots, and screen recordings of participants performing 3D modeling tasks. An example of a user-recorded screencast video can be seen in Figure 3.

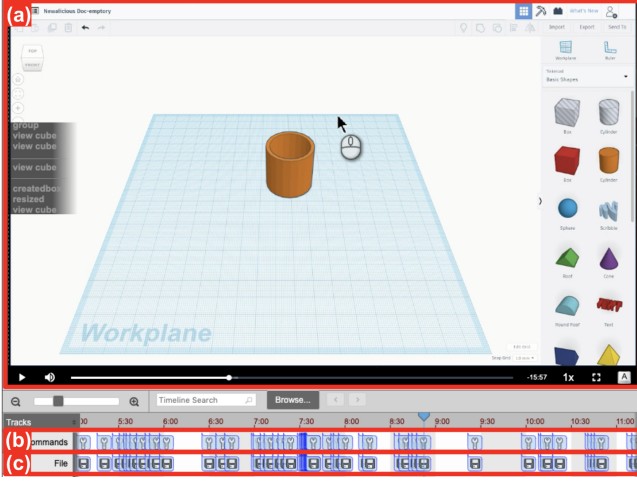

Figure 3: Screencast of a user demonstration, consisting of the (a) screen recording, (b) command sequences, and (c) 3D model snapshots

Using this approach, we collected user demonstrations for two tasks—modeling a mug and modeling a standing desk (Figure 4). These tasks were selected because they could be completed in under 30 minutes, and represent different levels of complexity. The mug task is relatively simple, requiring fewer operations and primitives, while the desk task can be complex and time consuming if the user does not have knowledge of particular Tinkercad tools, such as the Align and Ruler. The Desk model also requires approximately twice as many primitives as the Mug model.

We recruited participants through UserTesting.com and an email to an internal mailing list at a large software company. 14 participants were recruited for the Mug task, and 11 participants were recruited for the Desk task, but we excluded participants who did not follow the instructions, or failed to upload their recordings in the final step. After applying this criteria, we had 8 participants for the mug task (6 male, 2 female, ages 27–48), and 6 participants for the standing desk task (5 male, 1 female, ages 21–43).

The result of data collection procedure were 8 demonstrations for the Mug task, which took 26m:24s on average (SD=10m:46s) and consisted of an average of 142 command invocations (SD=101); and 6 demonstrations for the Desk task, which took 23m:23s on average (SD=8m:20s) and consisted of an average of 223 command invocations (SD=107).

### 4.2 Workflow to Graph Construction

The W-graphs construction pipeline consists of three steps: preprocessing, collapsing node sequences, and sequence merging.

[2]https://knowledge.autodesk.com/community/screencast

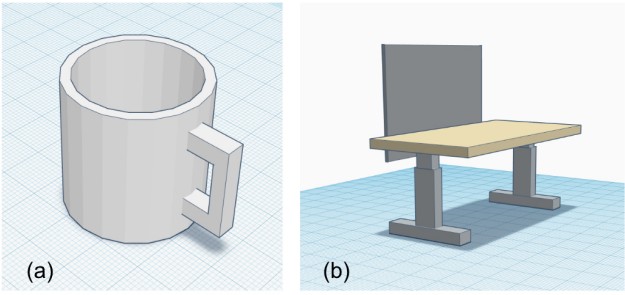

Figure 4: Models used for data collection – (a) Mug, (b) Desk

#### 4.2.1 Step 1. Preprocessing

To start, we collapse repeated or redundant commands in the sequence of events (both keystroke and clickstream data) for each demonstration. For example, multiple invocations of "arrow key presses" for moving an object are merged into one "object moved with keyboard" and multiple invocations of "panning viewpoint" are merged into "panning".

Next, the sequence of events for each user is considered as a set of nodes (one node per event), with directed edges connecting each event in timestamped sequence (Figure 5a). The 3D model snapshot for each event is associated with the corresponding node, and the event data (including timestamped command invocations) is associated with the incoming edge to that node. Since each demonstration starts from a blank document and finishes with the completed 3D model, we add a START node with directed edges to the first node in each demonstration, and we merge the final nodes of each demonstration into an END node. At this point, each demonstration represents a distinct directed path from the START node to the END node (Figure 5b).

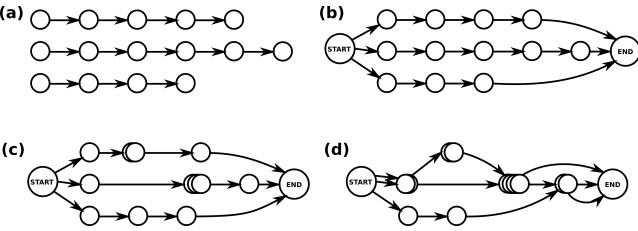

Figure 5: Illustration of how sequences get compressed and merged into a W-graph

#### 4.2.2 Step 2. Collapsing Node Sequences

Next, the pipeline merges sequences of nodes with similar geometry along each path from START to END, by clustering the snapshots of 3D model geometry associated with the nodes along each path (Figure 5c). The metric we use for 3D model similarity is discussed at the end of this section. To identify sequences with similar geometry, we first apply the DBSCAN [13] algorithm to cluster the 3D model snapshots associated with each path. We then merge contiguous subsequences of nodes that were assigned to the same cluster, keeping the 3D model snapshot of the final state in the subsequence as the representation of that node. We selected DBSCAN because it does not require a pre-defined number of clusters, as in alternative clustering algorithms such as K-Means. The hyperparameters of DBSCAN are tuned using the K-Nearest Neighborhood distance method, which is a standard practice for this algorithm [4, 5, 35].

### 4.2.3 Step 3. Sequence Merging

Finally, the pipeline detects "equivalent-intermediate" nodes across the paths representing multiple demonstrations (Figure 5d). To do this, we compute the 3D model similarity metric for all pairs of nodes that are not associated with the same demonstration (i.e., we only consider pairs of nodes from different demonstrations). We then merge all nodes with a similarity value below a threshold $\varepsilon$ that we manually tuned. In our experience, varying $\varepsilon$ can yield graphs that capture more or less granularity in variations in the task, and it would be interesting to consider an interactive system in which users can select a granularity that is suited to their use of the W-graph.

At this point, the W-graph construction is complete. As at the start of the pipeline, the directed edges from START to END collectively include all the events from the original demonstrations, but now certain edges contain multiple events (because the nodes between them have been collapsed), and some nodes are shared between multiple demonstrations.

### 4.3 Metrics for Detecting "Equivalent-intermediate States"

The most crucial part of the pipeline is determining the "similarity" between 3D model snapshots, as this is used to merge sequences of events in demonstrations, and to detect shared states across multiple demonstrations. We experimented with four different methods of computing similarity between 3D model snapshots, which we discuss below.

### 4.3.1 Comparing CSG trees

3D model snapshots are represented as CSG trees by Tinkercad, which consist of geometric primitives (e.g., cubes, cylinders, cones), combined together using Boolean operations (e.g., union, intersection, difference) in a hierarchical structure. A naive method of quantifying the difference between two snapshots would be to compare their respective trees directly, for example by trying to associate corresponding nodes, and then comparing the primitives or other characteristics of the tree. However, we quickly rejected this method because different procedures for modeling the same geometry can produce significantly different CSG trees. This makes the naive CSG comparison a poor method of judging similarity, where we specifically want to identify states where a similar end-result was reached through distinct methods.

### 4.3.2 Comparing 2D Images of Rendered Geometry

Inspired by prior work that has used visual summaries of code structure to understand the progress of students on programming problems [41], we next explored how visual renderings of the models could be used to facilitate comparison. We rendered the geometry of each 3D model snapshot from 20 different angles, and then compared the resulting images for pairs of models to quantify their difference. The appeal of this approach is that the method used to arrive at a model does not matter, so long as the resulting models look the same. However, we ultimately rejected this approach due to challenges with setting an appropriate threshold for judging two models as similar based on pixel differences between their renders.

### 4.3.3 Comparing 3D Meshes

Next, we experimented with using the Hausdorff distance [3], a commonly used mesh comparison metric, to compare the 3D meshes of pairs of 3D model snapshots. As with the comparison of rendered images, this method required extensive trial and error to set an appropriate threshold. However, the biggest drawback of this method was that the distances produced by the metric are in absolute terms, with the result that conceptually minor changes to a 3D model, such as adding a cube to the scene, can lead to huge changes in the distance metric. Ideally we would like to capture how "semantically"

meaningful changes are, which is not always reflected in how much of the resulting mesh has been altered.

### 4.3.4 Latent Space Embedding using Autoencoders

The final method we tried was to use an autoencoder to translate 3D point cloud data for each 3D model snapshot into a 512-dimensional vector. Autoencoders learn compact representations of input data by learning to encode a training set of data to a latent space of smaller dimensions, from which it can decode to the original data. We trained a latent model with a variation of PointNet [34] for encoding 3D point clouds to vectors, and PointSet Generation Network [14] for decoding vectors back to point clouds. The model was trained using the ShapeNet [43] dataset, which consists of 55 common object categories with about 51,300 unique 3D models. By using an additional clustering loss function [42], the resulting distributed representation captures the characteristics that matter for clustering tasks. One of the limitations of PointNet autoencoders is that current techniques cannot perform rotational-invariant comparisons of geometries. However, this fits nicely with our purpose, because rotating geometry does not affect semantic similarity for the 3D modeling tasks we are targeting.

Once trained, we can use the autoencoder to produce a 512-dimensional vector for each 3D model snapshot, and compare these using cosine distance to quantify the similarity between models. Overall, we found this to be the most effective method. Because it works using 3D point cloud data, it is not sensitive to how a model was produced, just its final geometry. Moreover, it required less tuning than comparing 2D images of rendered geometry or comparing 3D meshes, and in our experiments appeared to be more sensitive to semantically-meaningful changes to models.

### 4.4 Results

As a preliminary evaluation of the pipeline, we examined the graphs constructed for the mug and standing desk tasks. The W-graph for the mug task is shown in Figure 6. From the graph, a few things can be observed. First, the high-level method followed by most users was to first construct the body of the mug (as seen in paths A-B-C, and A-C), and then build and add the handle. Examining the screen recordings, all three users on path A-B-C created the body by first adding a solid cylinder and then adding a cylindrical "hole" object[3] to hollow out the center of the solid cylinder (see Figure 7a). Two of the three users on path A-C followed a slightly different method, creating two solid cylinders first, and then converting one of them into a hole object (Figure 7b). It is encouraging that the pipeline was able to capture these two distinct methods.

The remaining user on path A-C created a hole cylinder first, but ultimately deleted it and started again, following the same procedure as the users on path A-B-C. This highlights an interesting challenge in building W-graphs, which is how to handle backtracking or experimentation behavior (using commands such as Undo and Erase). We revisit this in the Discussion section at the end of the paper.

The users on paths A-D-E-F and A-E-F followed a different approach from those discussed above. Both of these users started by creating a cylinder (as a hole in the case of A-D-E-F, and as a solid in the case of A-E-F), then built the handle, and finally cut out the center of the mug's body. The A-D-E-F user built the handle through the use of a solid box and a hole box (Figure 8a), but the A-E-F user used a creative method—creating a primitive in the shape of a letter 'B', then cutting out part of it to create the handle (Figure 8b). Again, it is encouraging that the pipeline was able to separate these distinct methods.

For the modeling of the handle, nodes F, G, and H capture the behavior of building the handle apart from the body of the mug, and then attaching it in states I and J. The E-F transition seems strange

---

[3]Tinkercad shapes can be set as solid or as *holes*, which function like other shapes but cut out their volume when grouped with solid objects.

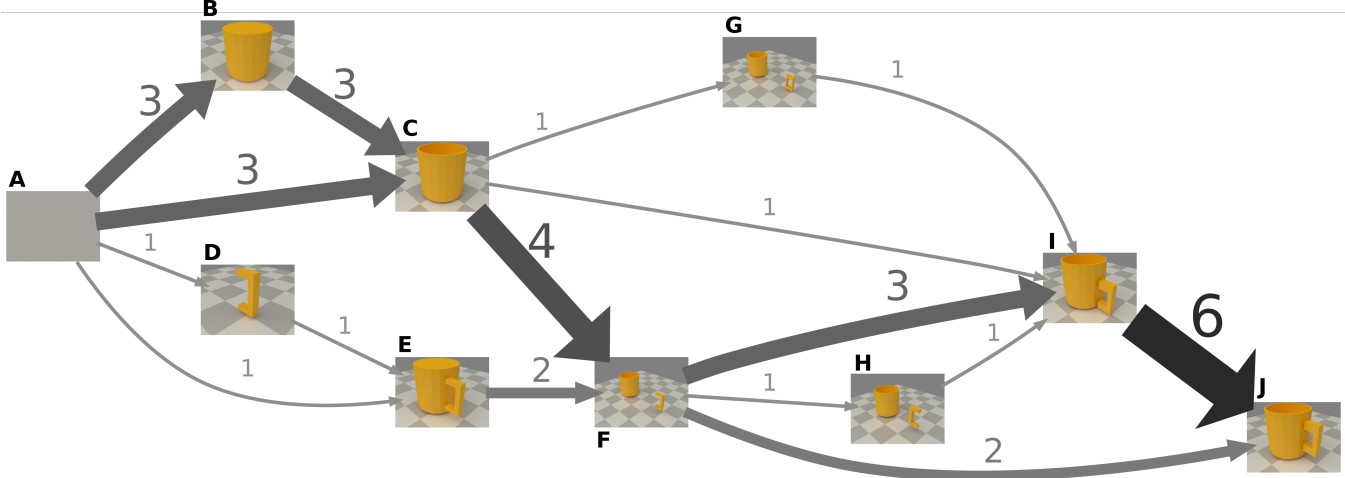

Figure 6: W-graph for the mug task. Edge labels indicate the number of demonstrations for each path. For nodes with multiple demonstrations, a rendering of the 3D model snapshot is shown for one of the demonstrations. A high-res version of this image is included in supplementary materials.

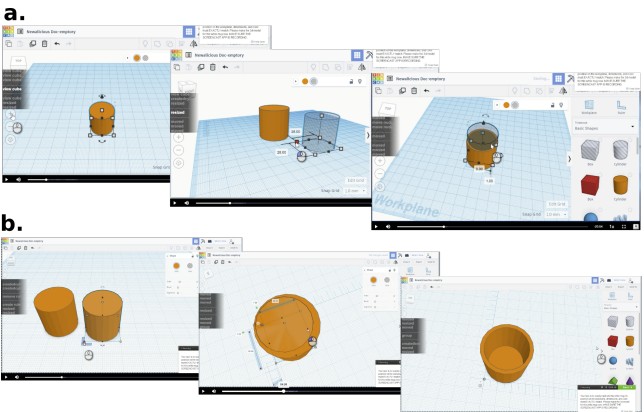

Figure 7: Two distinct methods of creating the mug body: (a) Create a solid cylinder, create a cylindrical hole, and group them; (b) Create two solid cylinders, position them correctly, then convert one into a hole.

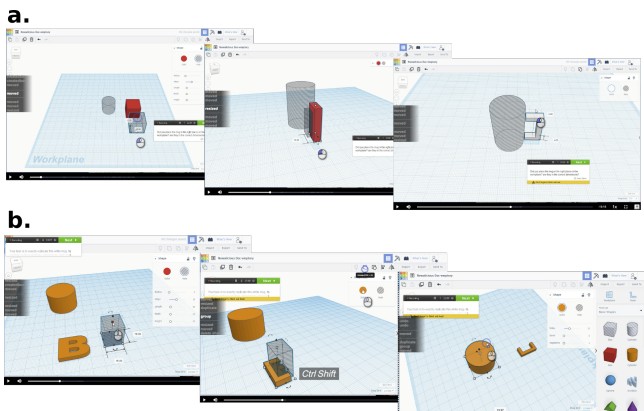

Figure 8: Two methods of creating the handle: (a) Combine a solid box and a box-shaped hole; (b) Cut a letter 'B' shape into the handle using several box-shaped holes.

in Figure 6, but reviewing the screen recording, the user moved the handle away from the mug before cutting the hole in the body, perhaps to create some space to work.

Overall, the pipeline appears to be effective in capturing the variety of methods used to create the body of the mug, and the edges of the graph captured a few distinct methods for creating the handle. An interesting observation is that the node identification algorithm did not capture any sub-steps involved in creating the handle. One possibility is that the methods used by different users were distinct enough that they did not have any equivalent-intermediate states until the handle was complete. Another possibility is that the autoencoder is not good at identifying similar states for models that are partially constructed (being trained on ShapeNet, which consists of complete models). The above having been said, this is not necessarily a problem as the edges do capture multiple methods of constructing the handle.

The W-graph for the standing desk task is shown in Figure 9. The graph is more complex than that for the mug task, reflecting the added complexity of creating the standing desk, but we do observe similarities in how the graph captures the task. In particular, we can see paths that reflect the different orders in which users created the three main parts of the desk (the top, the legs, and the privacy screen).

We also notice some early nodes with box shapes, which later diverge and become a desk top in some demonstrations, and legs in another. These nodes that represent a common geometric history for different final shapes are interesting, because they represent situations where the algorithm may correctly merge similar geometry, but doing so works counter to the goal of identifying workflows for completing sub-goals of the task, effectively breaking them up into several edges. A possible way to address this would be to modify the pipeline so it takes into account the eventual final placement of each primitive at the end of the task, or several edges forward, in determining which nodes to merge.

## 5  POTENTIAL APPLICATIONS OF W-GRAPHS

This section presents three novel applications that are made possible by W-graphs: 1) W-Suggest, a workflow suggestion interface, 2) W-Guide, an on-demand 3D modeling help interface, and 3) W-Instruct, an instructor dashboard for analyzing workflows.

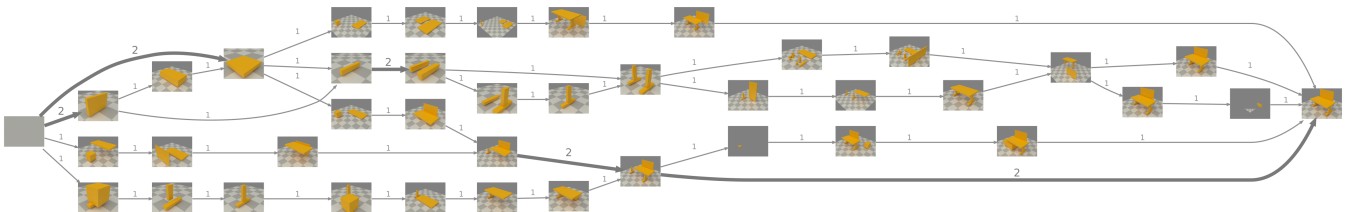

Figure 9: W-graph for the standing desk task. Edge labels indicate the number of demonstrations for each path. For nodes with multiple demonstrations, a rendering of the 3D model snapshot is shown for one of the demonstrations. A high-res version of this image is included in supplementary materials.

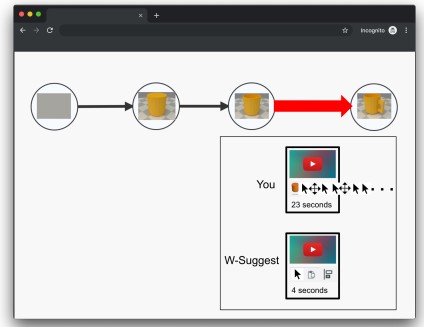

Figure 10: W-Suggest – A workflow suggestion interface mockup

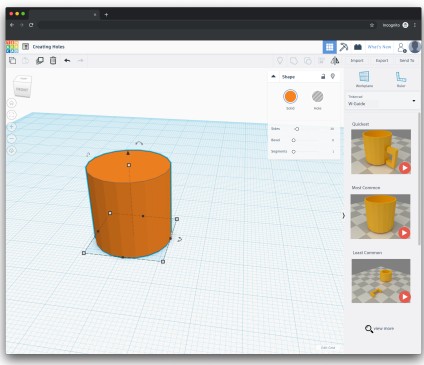

Figure 11: W-Guide – An on-demand task guidance interface mockup.

### 5.1 W-Suggest: Workflow Suggestion Interface

By representing the structure of how to perform a task, W-graphs can serve as a back-end for applications that suggest alternate workflows to users.

To use the W-Suggest system(Figure 10), the user first records themselves performing a 3D modeling task, similar to the procedure performed by participants in the previous section. However, instead of integrating this new workflow recording into the W-graph, the system compares the workflow to the existing graph and suggests alternate workflows for portions of the task.

W-Suggest uses the following algorithm to make its suggestions. First, it performs Steps 1 and 2 of the W-graph construction pipeline on the user's recording of the task (i.e., preprocessing the events, and collapsing node sequences with similar geometry). Next, the 512-dimensional embedding vector for each remaining 3D model snapshot is computed using the same autoencoder used for the W-graph construction pipeline. The vectors for each of these nodes are then compared to those of the W-graph nodes along the shortest path from START to END (as measured by total command invocations) to detect matches using the same threshold $\varepsilon$ used for graph construction. Finally, for each pair of matched nodes (one from the user, one from the shortest path in the W-graph), the edge originating at the user's node and the edge originating at the W-graph node are compared based on command invocations. Based on all of these comparisons, the algorithm selects the pair for which there is the biggest difference in command invocations between the user's demonstration and the demonstration from the W-graph. In effect, the idea is to identify segments of the user's task for which the W-graph includes a method that uses much fewer command invocations, which can then be suggested to the user.

### 5.2 W-Guide: On-Demand Task Guidance Interface

W-graphs could also serve as a back-end for a *W-Guide* interface that presents contextually appropriate video content to users on-demand as they work in an application, extending approaches taken by systems such as Ambient Help [32] and Pause-and-Play [33] with peer demonstrations.

While working on a 3D modeling task in Tinkercad, the user could invoke W-Guide to see possible next steps displayed in a panel to the right of the editor (Figure 11). These videos are populated based on the strategies captured from other users and stored in the W-graph. Specifically, the panel recommends video demonstrations from other users matched to the current user's state, and proceeds to the next "equivalent-intermediate" state (i.e., one edge forward in the graph). Using a similar approach to W-Suggest, these can be provided with meaningful labels (e.g., "Shortest workflow", "Most popular workflow", etc.).

W-Guide could use the identical algorithm as W-Suggest to construct a W-Graph and populate its suggestions. The only difference is that it would attempt to match the user's current incomplete workflow to the graph. This is achievable because the $\varepsilon$ threshold for collapsing node sequences is flexible, allowing W-Guide to construct a W-Graph from any point in current user's workflow and populate demonstrations for next steps.

An exciting possibility that becomes possible with W-Guide is that the system could dynamically elicit additional demonstrations from users in a targeted way (e.g., by popping up a message asking them to provide different demonstrations than those pre-populated in the panel). This could allow the system to take an active role in fleshing out a W-graph with diverse samples of methods.

### 5.3 W-Instruct: Instructor Tool

Finally, we envision the *W-Instruct* system in which W-graphs become a flexible and scalable tool for instructors to provide feedback to students, assess their work, and generate tutorials or other instructional materials on performing 3D modeling tasks.

W-Instruct (Figure 12) supports instructors in understanding the different methods used by their students to complete a task—by

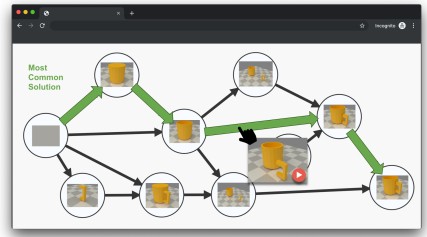

Figure 12: W-Instruct – An instructor tool mockup.

examining the graph, an instructor can see the approaches taken by students, rather than simply the final artifacts they produce. The grouping of multiple students' workflows could also be used as a means to provide feedback to a large number of learners at scale (e.g., in a MOOC setting). Also, the instructor could quickly identify shortcuts, crucial parts of the workflow to emphasize, or common mistakes by browsing the W-Graph. As shown in Figure 12, edges can be highlighted to show the most common solutions, and the video demonstration corresponding to an edge can be viewed by hovering over a node in the graph.

Along similar lines to W-Instruct, we see potential for W-graphs to support the generation of tutorials and other learning content, building on past work exploring approaches for generating tutorials by demonstration [10, 19]. For example, synthetic demonstrations of workflows could potentially be produced that combine the best segments of multiple demonstrations in the W-graph, creating a demonstration that is more personalized to the current user than any individual demonstration.

## 6  USER FEEDBACK ON W-SUGGEST

While the main focus of this work is on the computational approach for constructing W-graphs, we implemented the W-Suggest application as a preliminary demonstration of the feasibility of building applications on top of a constructed W-graph (Figure 13). The W-Suggest interface consists of a simplified representation of the user's workflow, with edges highlighted to indicate a part of the task for which the system is suggesting an improved workflow. Below this are two embedded video players, one showing the screen recording of the user's workflow for that part of the task, and the other showing a suggested workflow drawn from other users in the graph. Below this are some metrics on the two workflows, including duration, the distribution of commands used, and the specific sequences of commands used.

To gain some feedback on the prototype, we recruited 4 volunteers to perform one of the two tasks from the previous section (two for the mug task, two for the standing desk task) and presented them with their W-Suggest interface. We asked them to watch the two videos—one showing their workflow, the other showing the suggested workflow—and then asked a few short questions about the interface. Specifically, we asked if they felt it was useful to view the alternate demonstration, and why or why not they felt that way. We also asked them their thoughts on the general utility of this type of workflow suggestion system, and what aspects of workflows they would like suggestions on for software they frequently use.

Due to the small number of participants for these feedback sessions, they are best considered as providing preliminary feedback, and certainly not a rigorous evaluation. That being said, the feedback from participants was quite positive, with all participants agreeing it would be valuable to see alternative workflows. In particular, participants mentioned that it would be valuable to see common workflows, the fastest workflow, and workflows used by experts.

Two participants mentioned that they learned something new

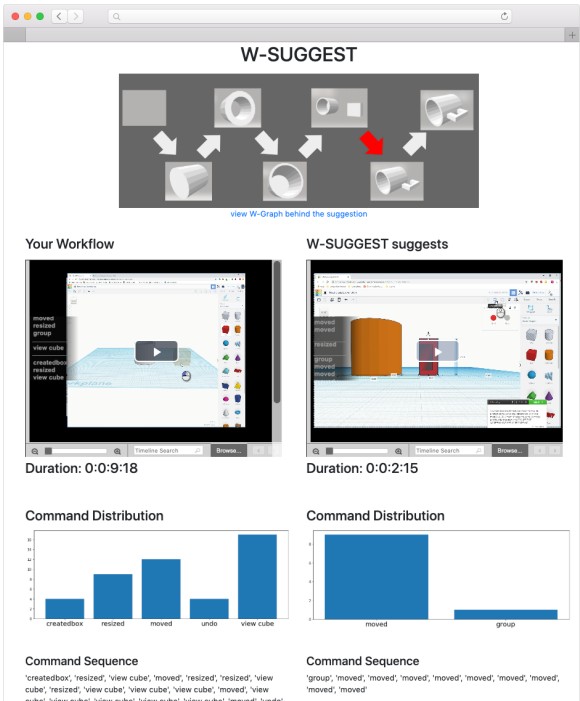

Figure 13: W-Suggest – The implemented interface.

about how to use Tinkercad from watching the alternate video, as in the following quote by P2 after seeing a use of the Ruler tool to align objects: *Oh, you can adjust the things there [with the Ruler] that's useful. Oh, there's like an alignment thing, that seems really easy.*

Likewise, P4 observed a use of the Workplane tool that he found valuable: *It's assigning relative positions with it [the Workplane and Ruler]—I wanted to do something like that.*

All participants agreed that efficiency is an important criterion when recommending alternative workflows. However, P1 and P2 noted that the best method to use in feature-rich software, or other domains such as programming, can often depend on contextual factors. In particular, P1 noted that they might prepare a 3D model differently if it is intended to be 3D printed. This suggests that additional meta-data on the users or the intended purpose for creating a model could be useful for making workflow recommendations.

## 7  DISCUSSION, LIMITATION AND FUTURE WORK

Overall, the W-graphs produced for the mug and standing desk tasks are encouraging, and suggest that our pipeline is effective at capturing different high-level methods for modeling 3D objects. Testing the pipeline on these sample tasks also revealed a number of potential directions for improving the approach, including modeling backtracking behavior in demonstrations, and accounting for sub-tasks with common intermediate states. Finally, our user feedback sessions for W-Suggest showed enthusiasm for applications built on W-graphs, and revealed insights into criteria for what makes a good demonstration, including the importance of contextual factors.

In this section we revisit the potential of modeling backtracking and experimentation, discuss the question of how many demonstrations are needed to build a useful W-graph graph, and suggest further refinements of the graph construction method. We then discuss how our approach could be generalized to building models of similar tasks.

## 7.1 Backtracking and Experimentation

In our current approach, Undo and Erase are treated the same as other commands. In some situations this may be appropriate, but at other times these commands may be used to backtrack, to recover from mistakes, or to try other workflows, and past work has shown their occurrence may indicate usability challenges [1]. It would be interesting to investigate whether these practices for using Undo and Erase could be detected and represented in a W-graph. This could take the form of edges that go back to previous states, creating directed cycles or self-loops in the graph. Applications built on top of a W-graph could also use the number of Undos as a metric for ranking paths through the graph (e.g., to identify instances of exploratory behavior), or as a filtering metric to cull the graph of such backtracking behavior.

## 7.2 Branching Factors and Graph Saturation

A nice feature of W-graphs is that they can be built with only a few demonstrations. As the number of demonstrations grows, the graph can more fully capture the space of potential workflows for the task. However, it is likely that the graph will eventually reach a point at which it is *saturated*, beyond which additional workflows will contribute a diminishing number of additional methods. The number of demonstrations needed to reach saturation will likely vary task by task, with more complex tasks requiring more demonstrations than simpler tasks. Examining how the sum of the branching factor for all nodes in the tree changes with each added demonstration may give an indication of when the graph has reached saturation, as the number of branches is likely to stop growing once new methods are no longer being added.

## 7.3 Scalability

In one sense, the W-graph approach is scalable by design, as it relies on computational comparisons of 3D models rather than human interventions such as expert labeling or crowdsourcing. However, more work is needed to understand how the structure of W-graphs produced by our pipeline change as the number of demonstrations in a graph grows. In particular, there is the question of how the parameters for identifying similar intermediate states may need to change in response to a growing number of workflows, in order to produce graphs at the right granularity for a given application, and other issues that may come up when processing many demonstrations. On the application end, metrics could be developed to identify less-used but valuable traces contained in a graph with many demonstrations.

## 7.4 Robustness Against Different Workflow Orders

A potential limitation of our current approach is that it preserves the global order of sub-tasks, including those that could be performed in an arbitrary order (e.g., a user could start by modeling the legs or the top of a table), and this could prevent it from grouping some variations of sub-tasks together if a given sub-task is performed first by some users, and later by others. Preserving the global order of sub-tasks has some advantages, in that it reveals how users commonly sequence the sub-tasks that make up the overall task, and it can also reveal cases where sub-tasks benefit from being ordered in a certain way, as may occur when objects built as part of a preceding sub-task are used to help with positioning or building objects in a subsequent sub-task. However, it would be interesting to look at approaches that post-process a W-graph to identify edges across the graph where the same sub-task is being performed (e.g., by looking for edges where similar changes to geometry are made, ignoring geometry that isn't changing) to address this limitation and gain insights into sub-task order in the graph.

## 7.5 Extension to Similar Tasks and Sub-Tasks

Another interesting direction for future work is to consider how the W-graph approach could be extended to scenarios where the demonstrations used to produce the graph are not of the exact same task, but instead represent workflows for a class of similar tasks (e.g., modeling chairs). We believe the autoencoder approach we have adopted could be valuable for this, as it is less sensitive to variations in the model, and potentially able to capture semantic similarities between models of different objects within a class, but more research is required. Sub-goal labels provided by users or learners could be valuable here, building on approaches that have been used for how-to videos [25] and math problems [40]. Given a user's explanation of their process or different stages in the task, the graph construction algorithm would have access to natural language descriptions in addition to interaction traces and content snapshots, which could be used to group workflows across distinct but related tasks.

Beyond refining our algorithms to work with similar tasks, it would be interesting to investigate how a large corpus of demonstrations could be mined to identify semantically-similar sub-tasks (which could be then turned into W-graphs). *Multi-W-graphs* could conceivably be developed that link together the nodes and edges of individual W-graphs, to represent similarities and relationships between the workflows used for different tasks. For example, nodes representing the legs of a desk, chair, or television stand could be linked across their respective graphs, and edges that represent workflows for creating certain effects (e.g., a particular curvature or geometry) could be linked as well. In the limit, one could imagine a set of linked graphs that collectively encode all the tasks commonly performed in a domain, and feed many downstream applications for workflow recommendation and improvement.

## 7.6 Generalizing to Other Software and Domains

Though we demonstrated our approach for 3D modeling software, the W-graph construction approach would be straightforward to extend to other software applications and domains. For many domains, such as 2D graphics or textual media, the technique could be generalized by simply substituting in an appropriate feature extraction mechanism for that domain. More challenging would be extending the approach to apply across a variety of software applications, perhaps by different software developers, where instrumentation to gather commands and content is not easy. To approach this, we could imagine using the screen recording data for the content, and accessibility APIs to gather the actions performed by users (an approach used in recent work [17]). Beyond fully-automated approaches, learnersourcing approaches [25] could be used to elicit sub-goals that have particular pedagogical value, and these peer-generated sub-goals could be turned into feedback for other learners in the system, using similar methods to those explored in other applications [24].

## 8 Conclusion

This work has contributed a conceptual approach for representing the different means by which a fixed goal can be achieved in feature rich software, based on recordings of user demonstrations, and has demonstrated a scalable pipeline for constructing such a representation for 3D modeling software. It has also presented a range of applications that could leverage this representation to support users in improving their skill sets over time. Overall, we see this work as a first step toward enabling a new generation of help and learning systems for feature-rich software, powered by data-driven models of tasks and workflows.

## 9 Acknowledgements

Thanks to Autodesk Research for all their support, and in particular to Aditya Sanghi and Kaveh Hassani, who provided invaluable advice and guidance on techniques for comparing 3D models. Thanks also to our study participants for their valuable feedback.

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
