# OpenReview forum: "Workflow Graphs: A Computational Model of Collective Task Strategies for 3D Design Software"
_graphicsinterface.org/Graphics_Interface/2020/Conference — GI 2020_

### Official Review · AnonReviewer1 · 2020-04-19
**Interesting idea and useful applications. Suggest accepting the paper**

**Rating:** 7
**Confidence:** 3

**Review:**

This paper presents a computational technique, W-graphs, that visualizes the different steps multiple people perform to complete the same task. The example task demonstrated in the paper is that of 3D modelling using TinkerCAD software. The contribution of the paper is threefold: an explanation for how W-graphs can be constructed, descriptions of potential applications for W-graph, and preliminary evaluation of W-Suggest (an application that suggests alternative workflows).

Overall, the paper is well written and presents an interesting idea. There is value in learning about how others perform the same task and, it can be useful to gain insights about the higher-level categories of sub-steps and the sequence in which people complete them to achieve the end result. The idea of representing workflows, especially in the context of software tool uses, isn't necessarily new, but the approach of synthesizing other people's sequences of higher-level actions is an interesting variant in my opinion.  The paper also did a good job explaining their explorations including alternative algorithms they considered for implementation.

While I liked the paper, there are a few questions I had:

I wondered if the potential benefit of W-graphs can be further refined. At a software level, there is a relatively low-cost associated with engaging in trial and error, so I wondered who would really benefit from seeing the sequence of high-level actions taken by others? For example, an expert may be more willing to try out a few options by themselves (and still not compromise on the time taken) without requiring to reference other people's workflows. Alternatively, if the W-graphs were more useful for novices, then is it enough to only know the sequence in which high-level actions were completed, or would they also need to know how the actual steps were performed?

I was also curious to know how the algorithm handled mistakes made by people. There is mention of treating erase and undo as one type of operation (mentioned in the future work), but it is unclear how they are currently represented in the W-graphs.

I wondered if the algorithm in any way distinguishes between useful information and those that are simply redundant or less valuable. The pre-processing stage seems to somewhat tackle this issue by collapsing reducant information, but I wondered if there was even a need to represent such information. For example, if a person rotated or used zoom several times to explore the model to view it from different angles, is that useful information for others to know about or is that simply information overload? I suppose that it depends on the application, but I wondered if the authors had considered some kind of data cleaning or filtering option.

Lastly, since the paper is primarily a technical algorithm contribution, it may help to provide more implementation details to enables others to implement or use the W-graphs in their applications. Perhaps the authors can consider sharing their code as an open-source resource?

Some minor comments:
There seem to be a few typos (E.g., SD=107 ?)

---

### Official Review · AnonReviewer2 · 2020-04-20
**Good idea, motivation could be strengthened.**

**Rating:** 7
**Confidence:** 3

**Review:**

The paper "Workflow Graphs: A Computational Model of Collective Task Strategies for 3D Design Software" presents the concept of workflow graphs (W-graphs) for representing multiple divergent solutions to 3D design tasks, a technique to generate these, and three suggestions for application areas.

Overall I enjoyed reading the paper. The paper is well written and presents a compelling idea. The authors do a good job covering related work and positioning their contribution to it. The three application examples present a good validation of the potentials of the idea presented.

That said, there are some things that I would like the authors to address:
 - The motivation of the paper is as it is weak, and could easily be strengthened as I believe the contribution as stronger potential than what is presented in the abstract and introduction. The abstract doesn't at all mention what motivated the work, and the introduction is vague and indirect. I would suggest the authors to emphasise the potential for computational support in 3D modeling tutorials as the main motivation.
 - It is unclear how well the idea scales beyond fixed examples. The authors address this a bit in the discussion, but I would have liked to see some technical details.
 - It isn't clear to me how Tinkercad was instrumented, as I assume everything didn't happen through Autodesk Screencast? How was, e.g., the 3D model snapshots made?
 - I would like the authors to discuss how labour intensive it is to create a w-graph where saturation is reached. Also, what is the increase in complexity and time when going from creating a w-graph for the mug task to the standing table? How much of this work can be crowd-sourced and how much requires expert supervision/curation?
 - The first sentence of "Workflow graphs" seems out of place in relation to the contribution as it is described in the introduction.

Overall I would recommend accepting the paper for GI 2020.

---

### Official Review · AnonReviewer3 · 2020-04-20
**Recommend for acceptance, but have some minor concerns regarding the related work and scalability issue.**

**Rating:** 9
**Confidence:** 4

**Review:**

# Summary
This paper presents W-graphs, a method to capture, classify, and simplify the multiple users' workflow of designing a 3D model. To construct the graph representation, the paper describes how to process the data and how to collapse and merge the similar nodes in the graph. Particularly, the autoencoder approach to determine the similarity of the 3D model in each edge seems effective and clever. I also appreciate the design rationale behind it. Based on this W-graphs engine, the authors suggest three possible applications and use scenarios.

# Review
Overall, I enjoyed reading this paper. I think the paper is well-motivated and well-written. It also reviews the literature well.
As the authors argue, I am also not aware of any prior work that investigates data-driven workflow analysis for 3D modeling tasks. Thus, I think the three contributions raised by the authors are valid, and these contributions should be strong enough. Therefore, I would like to recommend this for acceptance.

However, I have some comments or concerns that the authors should be able to address in revision.

# Suggestive user interfaces
Although the authors described three possible applications, these are essentially the same, in terms of providing the feature of "suggesting" the next step.
I think there are a lot of papers related to this suggestion feature in 3D modeling tasks.
For example,
- A Suggestive Interface for 3D Drawing
- Data-Driven Suggestions for Creativity Support in 3D Modeling
- Autocomplete 3D Sculpting
- Autocomplete Textures for 3D Printing
- Guided Exploration of Physically Valid Shapes for Furniture Design
- A Suggestive Interface for Image Guided 3D Sketching
(if we expand the application domain, I think there should be much more related work, such as Shadow Draw or Interactive Beautification.)

I still can see the novelty and benefits of the presented approach, as many of them provide suggestions based on the pre-defined algorithm or heuristics, but I also think there should be some more that use the data-driven approach for suggestion feature in 3D modeling (e.g., similar work as Chaudhuri and Koltun's Data-driven Suggestions).

Honestly, I think these works are more relevant to the current "Learning at Scale" of the Related Work. At least, the authors should expand the Discussion section about what are benefits and advantages, and what are and limitations or disadvantages, if any, when compared with these existing approaches.

# Scalability
I just wondered how this system can work in real-world scenarios. For example, in the paper's settings, the authors specifically confined the participants to model a mug or a desk. However, in the real world scenarios, the user would create a lot of variety of models, and the system need to learn and construct a graph based on all of them, if the system captures all of these workflows. Even in the simplest case, for example, if one provides the mixed data of both mug and desk workflows as an input, can the system still successfully construct a nice workflow-graph? Or, alternatively, do the authors think to collect every possible model in advance in the same approach (e.g., chair, airplane, lamp, etc)? It was not sure how it works in these scenarios. It is useful to clarify this point.

Also, it might be related to the above point (i.e., suggestive interfaces), but in the real-world W-suggest scenario, what the users want to create can be very diverse --- it is more likely that the user wants to create other than the mug. In that case, I was not sure how the proposed high-level workflow suggestion can work well. Obviously, in the presented user study, the participants were asked to create either a mug or a desk. However, it is not the case in real-world scenarios, particularly for professional users. Low-level feature suggestions (e.g., the suggestion of repetition, aligning, etc, similar to the related work listed above) can be generalizable for different use cases, but for the high-level feature suggestions proposed in this paper, I wonder if this lack of generalizability could become one of the disadvantages or not. I also want to hear the authors' opinions and ask the authors to clarify this point.

Given this point, I partly agree with the authors that this might be more useful for software learnability particularly for novice users (thus, I think the use of Tinkercad makes sense). I think it is fair for the authors to frame that this tool is intended for novices to learn how to model some pre-defined models. But, still, it is nice to see the discussion about the scalability and real-world deployment issue.

---

### Meta-Review · Area_Chair1 · 2020-04-22

**Recommendation:** Accept
**Confidence:** 4

**Metareview:**

Meta by R2:

Overall the reviewers were positive in their reviews, they all agree that the paper is well written and presents an interesting idea.
R1 finds the paper well-motivated an both R1 and R2 both point out that the paper has good coverage of related work.
As of improvements to the paper, R3 would like some more details on how mistakes are handled, and both R1 and R2 miss some technical details regarding the implementation. They also ask for more elaboration on the scalability of the approach.
In contrast to R1, R2 finds that the motivation could be strengthened in the introduction.

The paper is positively received by the reviewers who all recommend accepting it. My recommendation is therefore to accept the paper.
For a final version of the paper, the authors should carefully consider all reviewers' constructive comments.

---

### Decision · Program_Chairs · 2020-04-25

Accept